# Cognitive control of behavior and hippocampal information processing without medial prefrontal cortex

Eun Hye Park[1,2], Kally C O'Reilly Sparks[1†], Griffin Grubbs[1], David Taborga[1], Kyndall Nicholas[1], Armaan S Ahmed[1], Natalie Ruiz-Péreza[1], Natalie Kim[1], Simon Segura-Carrillo[1,3], André A Fenton[1,4*†]

[1]Center for Neural Science, New York University, New York, United States; [2]Psychiatry, Columbia University Irving Medical Center, New York State Psychiatric Institute, New York, United States; [3]Tandon School of Engineering, New York University, New York, United States; [4]Neuroscience Institute at the New York University Langone Medical Center, New York, United States

## eLife Assessment

This **important** study includes **convincing** evidence to show that behavioral measures and hippocampal representations when animals use task-relevant information and ignore irrelevant information do not depend on the medial prefrontal cortex. The results are expected to be of interest to those studying neural mechanisms of cognitive control and functions of associational brain regions.

*For correspondence:
afenton@nyu.edu

Present address: †Child and Adolescent Psychiatry, New York State Psychiatric Institute, New York, United States

**Abstract** Cognitive control tasks require using one class of information while ignoring competing classes of information. The central role of the medial prefrontal cortex (mPFC) in cognitive control is well established in the primate literature and largely accepted in the rodent literature because mPFC damage causes deficits in tasks that may require cognitive control, as inferred, typically from the task design. In prior work, we used an active place avoidance task where a rat or mouse on a rotating arena is required to avoid the stationary task-relevant locations of a mild shock and ignore the rotating task-irrelevant locations of those shocks. The task is impaired by hippocampal manipulations, and the discharge of hippocampal place cell populations judiciously alternates between representing stationary locations near the shock zone and rotating locations far from the shock zone, demonstrating cognitive control concurrently in behavior and the hippocampal representation of spatial information. Here, we test whether rat mPFC lesion impairs the active place avoidance task to evaluate two competing hypotheses, a 'central-computation' hypothesis that the mPFC is essential for the computations required for cognitive control and an alternative 'local-computation' hypothesis that other brain areas can perform the computations required for cognitive control, independent of mPFC. Ibotenic acid lesion of the mPFC was effective, damaging the cingulate, prelimbic, and infralimbic cortices. The lesion also altered the normal coordination of metabolic activity across remaining structures. The lesion did not impair learning to avoid the initial location of shock or long-term place avoidance memory, but impaired avoidance after the shock was relocated. The lesion also did not impair the alternation between task-relevant and task-irrelevant hippocampal representations of place information. These findings support the local-computation hypothesis that computations required for cognitive control can occur locally in brain networks independently of the mPFC.

## Introduction

Cognitive control is a psychological construct that refers to a set of processes that allow information processing and purposeful behavior to vary and adjust according to the demands of a subject's current goals. Without cognitive control, information processing and behavior would be rigid and inflexible when task demands and goals change (*Botvinick et al., 2001*). Because cognitive control is not limited to any one task or task domain, and because it engages various processes that include forms of attention, memory processes, behavioral inhibition, and the like, it has been a challenge to establish the neuronal basis of cognitive control. What would the neuronal network activity look like in a part of the brain that was implementing cognitive control? From our perspective, until we connect neuronal network function to cognitive control, the construct's value will be limited for understanding brain function and treating the diverse types of dysfunction that manifest as impaired cognitive control. Toward this goal, it has been important to identify regions of the brain that might be crucial for cognitive control. This has largely been approached by investigating brain function while subjects perform tasks designed to require cognitive control because performance depends on the ability to judiciously use task-relevant information while ignoring salient concurrent information that is currently irrelevant for the task.

Studies of human brain function identified subdivisions of the prefrontal cortex (PFC) that are engaged by tasks that are thought to require cognitive control, such as the Wisconsin Card Sort and Stroop tasks. As noted above, it is important to identify the neuronal network activity that the construct of cognitive control assumes must be operating. Recordings of neuronal activity from the PFC of non-human primates as they perform tasks that are thought to require cognitive control has been a particularly important development toward the identification of such neuronal network activity. Indeed, the consensus view is that PFC is crucial for cognitive control such that PFC activity is central to the underlying computations that operate through prefrontal interactions with diverse brain areas during cognitive control tasks (*Miller and Cohen, 2001*; *Guise and Shapiro, 2017*; *Marton et al., 2018*). An influential theory states: "We assume that the PFC serves a specific function in cognitive control: the active maintenance of patterns of activity that represent goals and the means to achieve them. They provide bias signals throughout much of the rest of the brain, affecting not only visual processes but also other sensory modalities, as well as systems responsible for response execution, memory retrieval, emotional evaluation, etc. The aggregate effect of these bias signals is to guide the flow of neural activity along pathways that establish the proper mappings between inputs, internal states, and outputs needed to perform a given task. This is especially important whenever stimuli are ambiguous (i.e. they activate more than one input representation), or when multiple responses are possible and the task-appropriate response must compete with stronger alternatives. From this perspective, the constellation of PFC biases—which resolves competition, guides activity along appropriate pathways, and establishes the mappings needed to perform the task—can be viewed as the neural implementation of attentional templates, rules, or goals, depending on the target of their biasing influence" (2, page 171).

Like biased competition, in which winner-take-all network representations compete to explain selective visual attention (*Spitzer et al., 1988*; *Moran and Desimone, 1985*; *Desimone and Duncan, 1995*; *Maunsell, 2015*; *Boudreau et al., 2006*) the theory is especially valuable and powerful because it predicts what neuronal signals should look like when cognitive control is operating. This enables experimentalists to operationally identify cognitive control using neuronal network behavior so that hypotheses can be rigorously evaluated. Accordingly, cognitive control would be at work when there is sustained neuronal network representations of task-relevant information that suppresses or gates representations of salient task-irrelevant information in accord with purposeful judicious behavior. This is not to say that the control signal itself would have this property, although a so-called 'transmissive' type of control signal would include task-relevant context and stimulus information, whereas a 'modulatory' type of control signal would only carry the control signal that biases expression of the task-relevant context and stimulus information (*Badre, 2024*; *Badre et al., 2021*).

The hippocampal role in the cognitive control of spatial and mnemonic information has been investigated using an active place avoidance task (*Chung et al., 2021*). The task conditions rodents to avoid the location of a mild shock and uses continuous rotation of the behavioral arena to dissociate the environment into two spatial frames, one defined by the rotating arena and the other defined by the stationary room (*van Dijk and Fenton, 2018*). Avoiding shock at a stationary room location

requires using the task-relevant spatial information and ignoring spatial information from the other, task-irrelevant arena frame. Dorsal hippocampus dysfunction impairs this avoidance without impairing the place avoidance when the two frames are not dissociated by stopping the rotation or attenuating the arena cues with shallow water (*Cimadevilla et al., 2000*; *Cimadevilla et al., 2001*; *Wesierska et al., 2005*; *Lee et al., 2012*). During active place avoidance task variants that require cognitive control, place cell discharge in hippocampus subfields, and head-direction cell discharge in the medial entorhinal cortical input to hippocampus demonstrate cognitive control of spatial representations; neural ensemble activity alternates between representing room and arena locations depending on the subject's proximity to the frame-specific location of shock (*Chung et al., 2021*; *van Dijk and Fenton, 2018*; *Kelemen and Fenton, 2010*; *Talbot et al., 2018*; *Park et al., 2019*).

A standard 'central-computation' hypothesis centralizes the computations that are necessary for cognitive control to the PFC and predicts that lesion of the PFC will impair active place avoidance (*Friedman and Robbins, 2022*; *Kamigaki, 2019*). Recent work indicating that the hippocampus (*Kelemen and Fenton, 2010*) and sensory thalamus (*Wimmer et al., 2015*) are also necessary for cognitive control can be reconciled with the central-computation hypothesis by assuming there are necessary interactions between modality-specific information processing in hippocampus or thalamus and the primary, control-specialized processing in the PFC (*Kamigaki, 2019*; *Murray et al., 2017*; *Preston and Eichenbaum, 2013*). Alternatively, the role of non-PFC brain structures in cognitive control could be explained by a 'local-computation' hypothesis; the computations needed for cognitive control can be performed locally in neural networks specialized for the particular information upon which the task depends and on which cognitive control operates. The local-computation hypothesis predicts that PFC lesion can spare cognitive control. We tested these mutually exclusive hypotheses by making medial prefrontal cortex (mPFC) lesions in rats and evaluating their behavior and hippocampal physiology during active place avoidance (*Figure 1A*). We reasoned that if the central-computation hypothesis was valid, active place avoidance behavior would be impaired by mPFC lesion along with discoordination of the alternating dynamics between concurrent hippocampal network representations of task-relevant room-frame and task-irrelevant arena-frame locations of the rotating arena. Alternatively, if the local-computation hypothesis was valid, then mPFC lesions would spare the active place avoidance behavior and the hippocampal place representation dynamics. We used cytochrome oxidase (CO), a metabolic marker of baseline neuronal activity, to confirm the mPFC lesions were effective and that there are non-local network consequences despite the local lesion. We first evaluated CO activity in regions known to be associated with performance in the active place avoidance task or regions with known connectivity to the mPFC. We then evaluated covariance of activity among the regions in an effort to detect network consequences of the lesion.

## Results

### mPFC lesion does not impair cognitive control in the active place avoidance task

Ibotenic acid, but not vehicle, targeted to mPFC caused lesions that included the cingulate cortex, prelimbic and infralimbic areas (*Figure 1B*; *Figure 1—figure supplement 1 and 2*). To assess the effect of mPFC lesion on behavior, we first examined locomotion during the pretraining sessions (*Figure 1C*). There is no effect of lesion on the total distance walked across the session (*Figure 1D*; $F_{1,16} = 0.04$, p=0.85, $\eta^2=0.002$). There is a main effect of trial ($F_{1,16} = 15.31$, p=0.001, $\eta^2=0.09$) but there is no effect of the group × trial interaction ($F_{1,16} = 0.14$, p=0.71, $\eta^2=10^{-3}$). Similarly, locomotion does not differ between the groups or across trials, nor is there a significant group × trial interaction during initial training on days 2–3 (df = 3.17/50.77) or during conflict training on days 4–5 (df = 4.97/79.58; $F'_S \leq 1.96$, p≥0.09, $\eta^2 \leq 0.03$).

We then examined the acquisition of active place avoidance memory during initial training using the time to first enter the shock zone, the clearest estimate of between-trial memory. Both groups learn to increase the latency to first enter the shock zone (*Figure 1E*). There is no effect of group ($F_{1,16} = 0.1$, p=0.76, $\eta^2=0.001$) but clear effects of day ($F_{1,16} = 69.61$, p=$10^{-7}$, $\eta^2=0.17$) and trial ($F_{5.33, 85.33} = 12.15$, p=$10^{-9}$, $\eta^2=0.12$). There are also no significant interactions (group × day: $F_{1,16} = 0.94$; p=0.35, $\eta^2=10^{-3}$; group × trial: $F_{5.33, 85.33} = 1.83$, p=0.11, $\eta^2=0.02$; day × trial: $F_{4.78, 76.43} = 1.93$, p=0.10, $\eta^2=0.03$; group x day × trial: $F_{4.78, 76.43} = 0.26$, p=0.93, $\eta^2=10^{-3}$). Neither was there a difference in

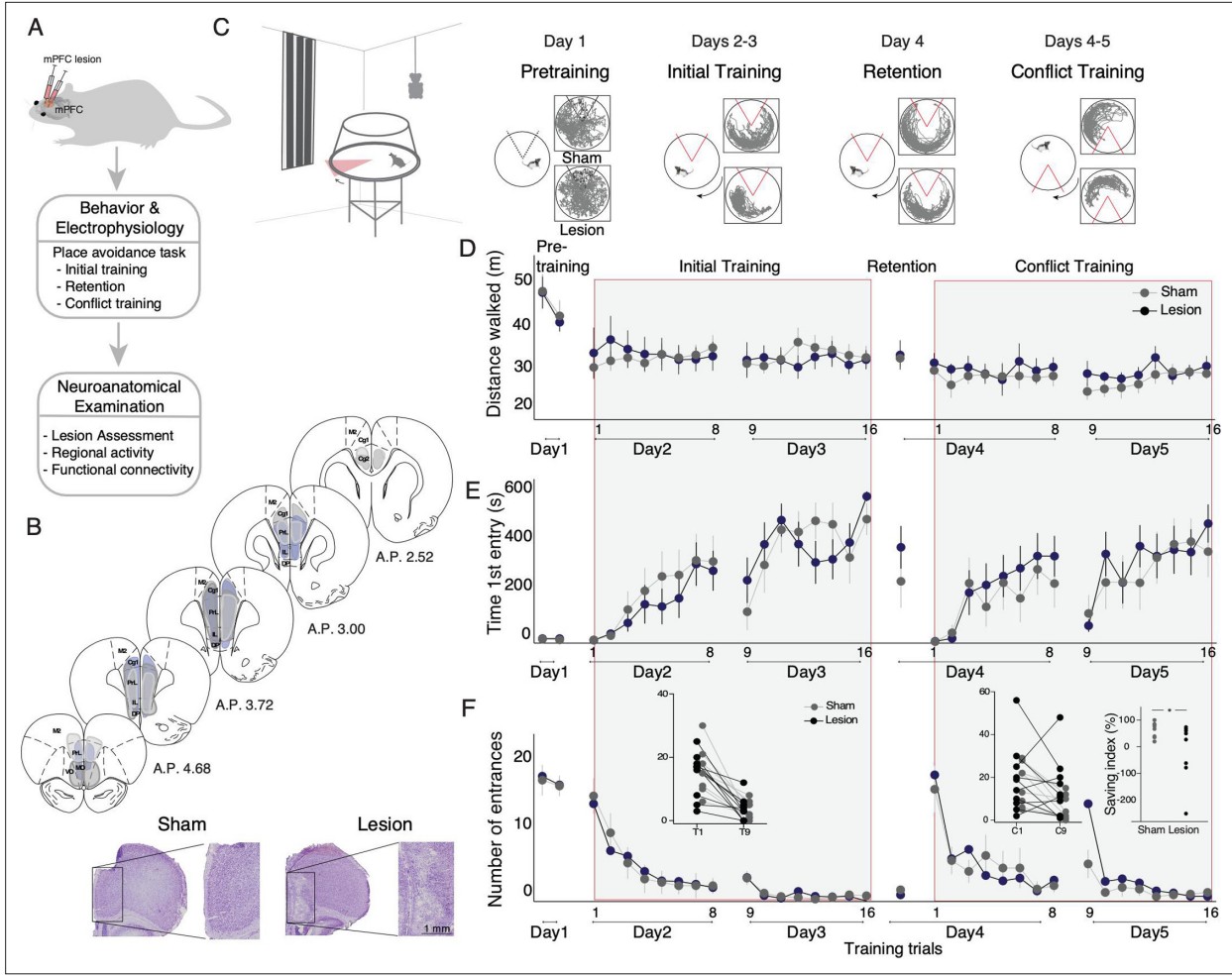

**Figure 1.** Medial prefrontal cortex (mPFC) lesion does not impair initial active place avoidance learning, but impairs cognitive flexibility in the conflict task variant. (**A**) Workflow to assess the impact of mPFC or sham lesions on spatial cognitive control. (**B**) Assessment and impact of the mPFC lesion in three representative subjects (light gray [intermediate with dorsal anterior lesion], dark gray [intermediate with ventral anterior lesion – includes medial orbital cortex], and purple [largest lesion]). The smallest lesion spanned A-P 3.24–2.52. PrL (prelimbic), IL (infralimbic), Cg (cingulate cortex), M2 (secondary motor cortex). Representative mPFC from a sham and a lesion rat (bottom). (**C**) Tracked room-frame positions from two example rats across active place avoidance training. The shock zone is indicated as a 60° sector, gray (shock off) and red (shock on), and arena rotation by the curved arrow. Day 1– pretraining: free exploration with shock off; days 2 and 3 – initial training: eight daily trials to avoid the shock zone. Day 4 – retention: one trial with shock on. Days 4 and 5 – conflict training: eight daily trials to avoid the shock zone relocated 180° from the initial location. Sham and mPFC lesion rats did not differ in (**D**) locomotor activity, (**E**) avoidance memory, or (**F**) place learning. (**F**) The left inset compares the number of entrances during the first 5 min of the first (D2T1) and second days of initial training (D3T9). The right inset compares the first 5 min of the first (D4C1) and second (D5C9) days of conflict trials. The percent difference in number of entrances between D5C9 and D4C1 of conflict training was computed as a savings index. Savings was not related to lesion size $r$=0.009, p=0.98. *p<0.05. Sham: n=8; lesion: n=10.

The online version of this article includes the following figure supplement(s) for figure 1:

**Figure supplement 1.** Targeting the medial prefrontal cortex (mPFC) injections.

**Figure supplement 2.** Medial prefrontal cortex (mPFC) lesions.

avoidance learning between the groups measured by the number of entrances into the shock zone (*Figure 1F*). The effect of group is not significant ($F_{1,16}$ = 0.002, p=0.96, $\eta^2$=10$^{-5}$) but the effects of day ($F_{1,16}$ = 26.34, p=10$^{-3}$, $\eta^2$=0.19) and trials ($F_{3.22, 51.50}$ = 36.76, p=10$^{-8}$, $\eta^2$=0.1) are significant. There are no significant interactions (group × day: $F_{1,16}$ = 0.003, p=0.96, $\eta^2$=10$^{-5}$; group × trial: $F_{3.22, 11.50}$ = 0.74, p=0.54, $\eta^2$=10$^{-3}$; group × day × trial: $F_{4.05, 64.81}$ = 0.35, p=0.74, $\eta^2$=10$^{-3}$). But the interaction between day and trial is significant (day × trial: $F_{2.26, 36.08}$ = 13.75, p=10$^{-5}$, $\eta^2$=0.11). The two groups are also indistinguishable on day 4 during which retention of 24 h memory is assessed by their times

to first enter the shock zone ($t_{16}$=1.01, p=0.3, d=0.49) and the number of entrances ($t_{16}$=0.95, p=0.4, d=0.002).

To increase the cognitive challenge, we were motivated by evidence that mPFC manipulation can affect cognitive flexibility as assessed by reversal learning tasks (*Ragozzino et al., 1999*; *McDonald et al., 2007*). Accordingly, we trained the rats to avoid the shock zone after relocating it 180° to assess the impact of mPFC lesion on cognitive flexibility with the additional challenge to distinguish between the current and previously learned location of shock, under the cognitive control challenge (*Rich and Shapiro, 2007*). Both groups learn the conflict task variant, measured by the number of entrances. There is no effect of group ($F_{1,16}$ = 0.21, p=0.65, $\eta^2$=10$^{-3}$), but the effects of day ($F_{1,16}$ = 8.99, p=0.01, $\eta^2$=0.05) and trials ($F_{1.89, 30.12}$ = 18.55, p=10$^{-9}$, $\eta^2$=0.28) are significant. The group interactions are not significant (group × day: $F_{1,16}$ = 0.55, p=0.47, $\eta^2$=10$^{-3}$; group × trial: $F_{1.88, 30.12}$ = 1.38, p=0.27, $\eta^2$=0.02; day × trial: $F_{2.30, 36.87}$ = 2.13, p=0.11, $\eta^2$=0.02, group × day × trial: $F_{2.30, 36.87}$ = 1.16, p=0.33, $\eta^2$=10$^{-3}$). Also, the time to enter the new shock zone, which was low in both groups, consistent with poor between-day memory for the novel location of shock (*Figure 1*, days 4–5 conflict training). There is no effect of group ($F_{1,16}$ = 0.21, p=0.65, $\eta^2$=10$^{-3}$), but the effects of day ($F_{1,16}$ = 20.46, p=0.01, $\eta^2$=0.04) and trials ($F_{1.89, 30.12}$ = 11.52, p=10$^{-3}$, $\eta^2$=0.13) are significant. The group interactions are not significant (group × day: $F_{1,16}$ = 0.03, p=0.86, $\eta^2$=10$^{-3}$; group × trial: $F_{1.88, 30.12}$ = 0.78, p=0.54, $\eta^2$=0.01; group × day × trial: $F_{2.30, 36.87}$ = 0.47, p=0.76, $\eta^2$=10$^{-2}$). Also, the interaction between day and trial is not significant (day × trial: $F_{2.30, 36.87}$ = 1.29, p=0.28 $\eta^2$=0.02). We more closely examined conflict learning by analyzing the initial experience of the new shock zone location when the rats are first confronted with the change in the location of shock. We compared the number of entrances during the first 5 min on the first trial of each day to estimate savings, measured as each animal's difference in performance during initial training (*Figure 1F*, left inset) and during conflict training (*Figure 1F* middle inset). During initial training, both sham (paired $t_7$=4.5, p=0.003, d=1.2) and lesion (paired $t_9$=4.7, p=0.001, d=1.5) rats improved. During conflict training, sham rats improved across the pair of first conflict trials of each day (*Figure 1F*, middle inset; paired $t_7$=3.8, p=0.006, d=1.4) but not lesion rats (paired $t_9$=0.87, p=0.4, d=0.27). Indeed, while all sham rats improved across the first two conflict trials of each day only 6/10 lesion rats improved (test of proportions z=2.3, p=0.03). We also quantified the improvement for each subject during conflict as a savings index (*Figure 1F*, right inset). The sham rats improved, avoiding more than the lesion rats in the first 5 min of the second conflict day compared to the first conflict day ($t_{16}$=2.4, one-tailed p=0.03, d=0.85). This difference on the second day of conflict training was observed despite equivalent performance across eight trials on the first day. We therefore

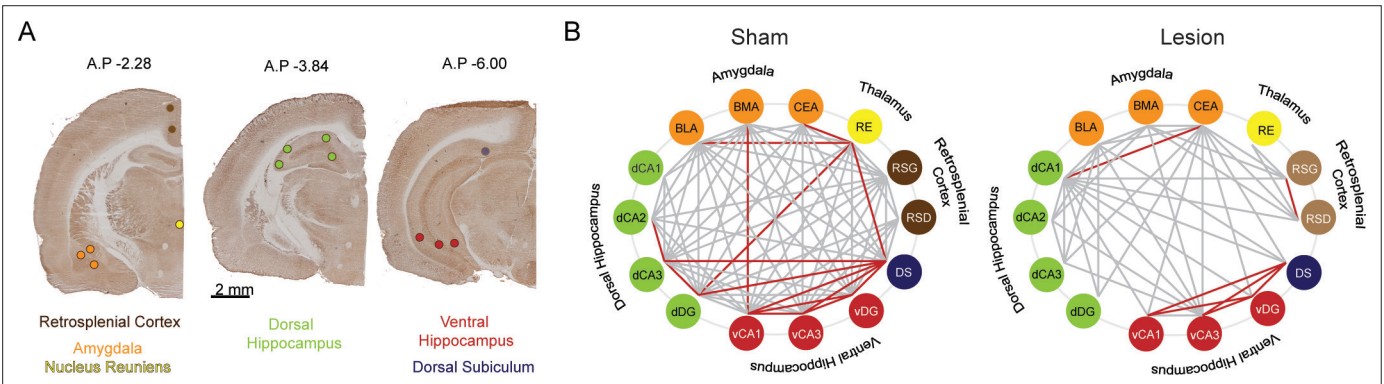

**Figure 2.** Cytochrome oxidase (CO) analysis demonstrates widespread metabolic consequences of medial prefrontal cortex (mPFC) lesion. (**A**) Representative CO staining and locations of the optical density readings. CO activity was measured in the dysgranular and granular retrosplenial cortices (RSD and RSG, respectively), the nucleus reuniens (RE), the central nucleus of the amygdala (CEA), basomedial and basolateral amygdala (BMA and BLA, respectively), the dorsal hippocampus CA1, CA2, CA3 and dentate gyrus areas (dCA1, dCA2, dCA3, dDG, respectively) the ventral hippocampus CA1, CA3 and dentate gyrus areas (vCA1, vCA3, vDG, respectively), and the dorsal subiculum (DS) marked as colored circle areas. (**B**) Interarea covariations of CO activity by graph theory analysis. Each line indicates a significant correlation (p<0.05) between the two brain regions ('nodes') it connects; only the red lines survived false discovery rate (FDR <0.01) correction. Sham: n=8; lesion: n=8.

The online version of this article includes the following figure supplement(s) for figure 2:

**Figure supplement 1.** Medial prefrontal cortex (mPFC) lesions change covarying resting-state metabolic activity between the dorsal and ventral hippocampus.

find no evidence that mPFC lesion causes a deficit in active place avoidance learning to avoid the initial location of shock, nor do we observe a deficit in memory retention up to 24 h for the initial shock location, both of which require cognitive control. However, the lesion is sufficient to impair conflict learning after a 24 h delay.

## mPFC lesion alters functional relationships amongst related brain areas

CO, a sensitive metabolic marker for neuronal function (*Wong-Riley, 1989*), was used to evaluate whether lesion effects were restricted to the mPFC. In a subset of rats (lesion n=8, sham n=8), CO activity was evaluated in 14 functionally-related brain regions (*Figure 2A*). CO activity in the central amygdala, but not elsewhere, is altered by the lesion, with activity increased in lesion rats (*Supplementary file 1*).

Pair-wise interregional correlations of CO activity were evaluated to assess whether extra-mPFC functional relationships are altered after mPFC lesion. After FDR correction (0.01), 15 of the 91 correlations are significant in the sham sample but only 7 in the lesion sample (*Figure 2B*, *Figure 2—figure supplement 1B*; test of proportions: z=2.26, p=0.03). The correlations within the ventral hippocampus are preserved after mPFC lesion, but some correlations are lost within the dorsal hippocampal formation (*Figure 2B*). Correlations between the nucleus reuniens and both the dorsal and ventral hippocampus are also lost after mPFC lesion. Correlations between the basolateral amygdala and nucleus reuniens and between the basomedial amygdala and ventral hippocampus are lost after mPFC lesion, and a correlation between dorsal CA1 and the central nucleus of the amygdala appears after mPFC lesion. These data indicate that the mPFC lesion causes functional changes beyond mPFC.

## mPFC lesion does not alter cognitive control of hippocampal neural representations

When the discharge of principal cells of the hippocampus is location-specific to so-called place fields, the cells are called place cells. Place fields are disrupted by molecular manipulations that disrupt spatial learning and memory and changes in place cell firing tend to track environmental changes like geometry and other spatial features that often accompany spatial learning and memory challenges (review *Jeffery and Hayman, 2004*). In contrast, spatial learning and memory tasks that depend crucially on hippocampus and hippocampal synaptic plasticity, like active place avoidance or task changes, such as those introduced by the conflict task variant, cannot be assessed by simple changes in place fields and their quality (*Chung et al., 2021*; *van Dijk and Fenton, 2018*; *Talbot et al., 2018*; *Fenton, 2024*; *Jeffery et al., 2003*; *Levy et al., 2019*; *Pavlowsky et al., 2017*; *Park et al., 2015*; *Pastalkova et al., 2006*). This is in part because dynamic cognitive variables, like adjustments of attention and cognitive control, operate during learning and memory tasks. The dynamics of such internal cognitive variables modulate place cell firing, creating extra-positional signals (*Chung et al., 2021*; *Kelemen and Fenton, 2010*; *Talbot et al., 2018*; *Kelemen and Fenton, 2013*; *Fenton et al., 2010*; *Kelemen and Fenton, 2016*). The presence of these extra-positional signals can be detected as momentary discharge deviations from the expectations of firing-field-based spatial discharge (*Johnson et al., 2009*). On the timescale of the few seconds it takes to walk across a firing field, these noise-like deviations can be measured as overdispersion (*Olypher et al., 2002*; *Olypher et al., 2003*). We can also measure the momentary positional information in the collective neuronal population discharge dynamics, which during active place avoidance alternates between encoding locations in the room and locations in the arena (*Kelemen and Fenton, 2010*; *Olypher et al., 2003*). The hippocampal and entorhinal cortex neuronal populations, each fluctuate between representing space in these two distinct spatial frames every few seconds (*Park et al., 2019*; *Fenton, 2024*; *Levy et al., 2023*). These representational dynamics are purposeful and can be summarized as a spatial frame ensemble preference (SFEP) that is biased to room-frame locations near the room-defined shock zone and is biased to arena-frame locations far from the shock zone (*Chung et al., 2021*; *van Dijk and Fenton, 2018*).

In light of these cognitive dynamics in neuronal activity representations of position and the ability to decode task information from neuronal population discharge, it is powerful to characterize cognitive control by investigating neuronal population discharge. Cognitive control requires the subject to process and use a class of information purposefully at the expense of other competing information. Accordingly, we previously demonstrated such a cognitive control signal in the population dynamics of dorsal hippocampus spatial discharge representations of room and arena positions as rats and mice

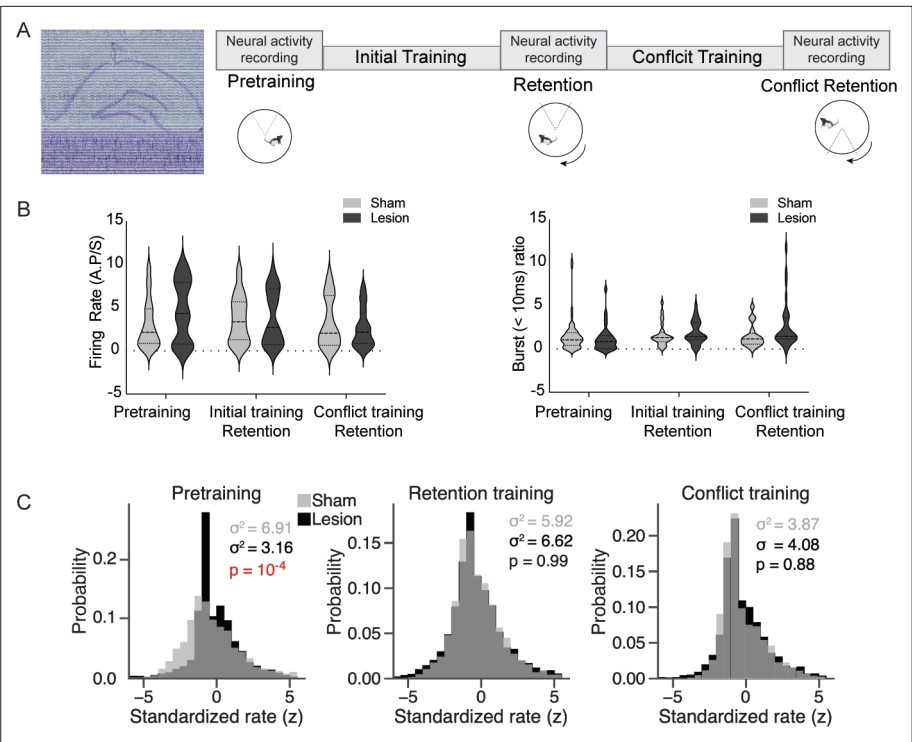

**Figure 3.** Medial prefrontal cortex (mPFC) lesion does not change basic discharge properties, but decreases hippocampus place cell overdispersion only in the absence of the cognitive control challenge. (**A**) Left: representative histology with overlaid recording traces from the Neuropixel probe. Right: recording schedule workflow during the cognitive control task. (**B**) There is no difference between sham and mPFC lesion rats in firing rate (sham: 3.29±0.27, lesion: 3.68±0.33, $t_{184}$=0.91, p=0.36) and burst ratio in hippocampal neurons (sham: 1.53±0.25, lesion: 2.02±0.28, $t_{174}$=1.29, p=0.19). (**C**) Distribution of standardized place cell discharge (z scores) computed during every 5 s episode in which the rat passed through place cell firing fields in the data set. Different numbers of passes qualified for evaluation during the pretraining (sham = 2777, lesion = 3398), retention (sham = 4039, lesion = 3366), and conflict (sham = 1748, lesion = 2336) recordings. The variance of the histograms characterizes overdispersion, statistically evaluated by their ratio as an F-test (pretraining: $F_{2776,3397}$ = 2.19, p=5.8 × $10^{-4}$). Sham: n=3; lesion: n=3 rats.

navigate on a rotating arena (*Chung et al., 2021*; *van Dijk and Fenton, 2018*; *Kelemen and Fenton, 2010*; *Kelemen and Fenton, 2016*). Consequently, we examined the SFEP, the representational dynamics of hippocampus CA1 discharge on the rotating arena during pretraining, as well as retention of the conditioned place avoidance of the initial and conflicting shock zone locations (*Figure 3A*). Note that the physical conditions were identical in these three trials because the shock was off.

We first examined basic discharge properties of individual CA1 principal cells, which do not differ between the sham and mPFC lesion groups (*Figure 3B*, *Supplementary file 2*). Nor does mPFC lesion alter thalamic firing rates, but it reduces the likelihood of bursting activity (*Supplementary file 3*) complementing the CO evidence that the mPFC lesion affected activity in areas beyond the lesion sites.

We then examined overdispersion of place cell firing, which is known to be reduced by prefrontal inactivation (*Hok et al., 2013*). In contrast, overdispersion is expected to increase with cognitive control and other extra-positional processes that increase discharge non-stationarity (*Fenton et al., 2010*). We find that during pretraining overdispersion of hippocampal discharge is reduced by a factor of two in lesion compared to sham rats, consistent with an effective lesion (*Figure 3C,*, left). Might this also indicate reduced cognitive control in the lesion rats? Remarkably, during conditioned place avoidance the overdispersion in lesion rats increases to the level observed in sham rats, consistent with the increased demand for cognitive control (*Figure 3C*, middle and right). These findings confirm the effectiveness of the prefrontal lesion and also provide electrophysiological evidence consistent with intact cognitive control in the lesion rats.

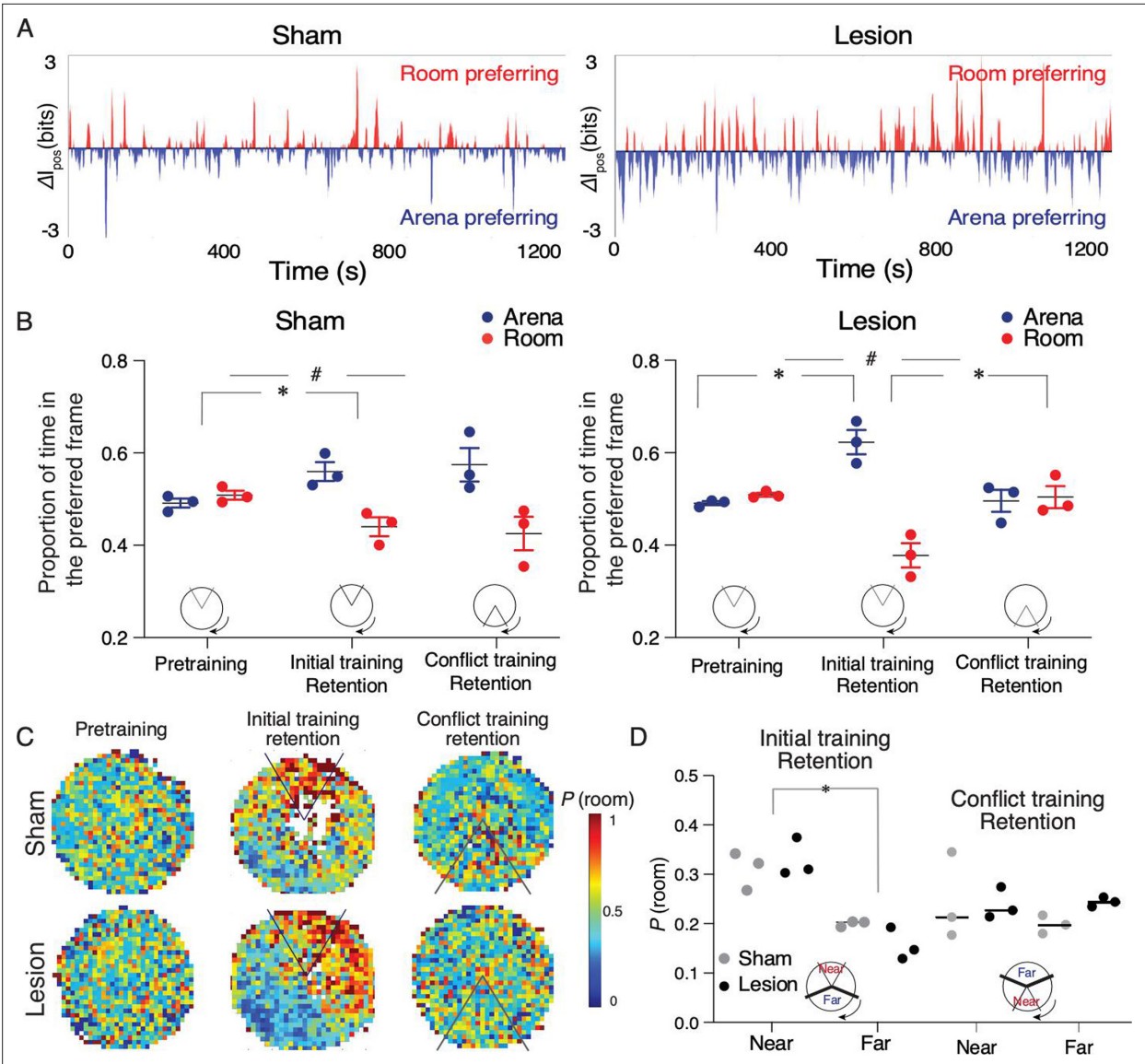

**Figure 4.** Sham and medial prefrontal cortex (mPFC) lesion rats do not differ in expressing cognitive control of spatial frame-specific representations of location. (**A**) Individual ensemble examples during the day 4 retention session and (**B**) group statistics of spatial frame ensemble preference (SFEP), demonstrating cognitive control in both groups. During retention, SFEP is biased to the arena frame in both the sham ($t_4$=4.16, #p=0.01) and lesion rats ($t_4$=6.62, #p=0.003). *p<0.05 post-hoc differences. (**C**) Group spatial probability distribution of SFEP for room frame preference during the pretraining, initial, and conflict retention sessions with no shock. (**D**) Summary of average probability of room-preferring SFEP discharge in half the arena near and far from the shock zone during the initial and conflict retention sessions. Two-way group × location ANOVA (sham: n=3, lesion: n=3) during retention of the initial shock zone location: Group: $F_{1,4}$ = 0.17, p=0.68, $\eta^2$=10$^{-3}$; location: $F_{1,4}$ = 54.69, p=0.001, $\eta^2$=0.84; group × location $F_{1,4}$ = 2.04, p=0.2, $\eta^2$=10$^{-2}$; during retention of the conflict shock zone location: group: $F_{1,4}$ = 0.71, p=0.45, $\eta^2$=0.05; location: $F_{1,4}$ = 0.42, p=0.55, $\eta^2$=0.056; group × location $F_{1,4}$ = 0.67, p=0.45, $\eta^2$=0.08. Sham: n=3; lesion: n=3 rats.

We next examined the SFEP, the electrophysiological signature of cognitive control of place representations, during conditioned place avoidance. CA1 ensemble discharge alternates between preferentially representing room locations and arena locations (***Figure 4A***; runs tests for all recordings were significant; z's≥12.3, p's≤10$^{-33}$). During pretraining, SFEP is equally likely to signal the current room location or alternatively, the current arena location, in both the sham and lesion rats (***Figure 4A***). Importantly, during avoidance of the initial shock location, SFEP favors the arena frame because the rat avoids the room frame location of shock (***Figure 4B***) and SFEP is room-preferring near the shock zone and arena-preferring far from the shock zone (***Figure 4C and D***), as previously demonstrated (***Chung et al., 2021***; ***van Dijk and Fenton, 2018***). Relocating the shock zone 180° for conflict training

changes SFEP, but only in the lesion group such that it is equally probable to be room- or arena-preferring, consistent with needing to decide between avoiding the current and previous shock location throughout the environment (*Figure 4B*; *Dvorak et al., 2018*). The ability to manipulate SFEP of hippocampal representational dynamics by the presence and location of shock directly demonstrates cognitive control at the level of the hippocampus. Despite an influence of mPFC lesion on cognitive control and the non-stationarity of hippocampal representations of space in hippocampal discharge, the lesion does not impair cognitive control of place representations in the place avoidance task.

## Discussion

Although permanent lesion of the mPFC alters baseline metabolic coupling of mPFC- and hippocampus-related brain areas (*Figure 2*, *Figure 2—figure supplement 1*), as well as alters hippocampal representational dynamics during expression of cognitive flexibility in an active place avoidance task (*Figure 3*), we find no behavioral or electrophysiological evidence that the lesion impairs cognitive control in the basic active place avoidance task (*Figures 1, 3 and 4*), despite the lesion being sufficient to cause a cognitive flexibility deficit when the shock zone was relocated for the conflict training trials (*Figure 1F*, insets). We observed increased overdispersion in lesion rats but only during pretraining; place cell tuning was not disturbed, only the discharge reliability was reduced in the absence of the place task. We also observed reduced bursting in the discharge of the underlying posterior and lateral posterior thalamus in the lesion group compared to sham (*Supplementary file 3*). Thus, CO imaging and electrophysiological evidence identify changes in the brain beyond the directly damaged mPFC area. In particular, the dorsal hippocampus loses the inhibitory input from mPFC (*Hoover and Vertes, 2007*; *Malik et al., 2022*) and loses the metabolic correlation with the nucleus reuniens, which is thought to be a relay between the mPFC and the dorsal hippocampus (*Griffin, 2015*; *Vertes et al., 2006*).

We have previously demonstrated cognitive control in the active place avoidance task variant we used (*Figure 1*) because the rats must ignore local rotating place cues to avoid the stationary shock zone. Even when the arena does not rotate, rats distinctly learn to avoid the location of shock according to distal visual room cues and local olfactory arena cues, such that the distinct place memories can be independently manipulated using probe trials (*Fenton et al., 1998*; *Bures et al., 1997*). When the arena rotates, as in the present studies, neural manipulations that impair the place avoidance are no longer impairing when the irrelevant arena cues are hidden by shallow water (*Wesierska et al., 2005*; *Lee et al., 2012*; *Kao et al., 2010*; *Park et al., 2024*). Furthermore, persistent hippocampal neural circuit changes caused by active place avoidance training are not detected when shallow water hides the irrelevant arena cues to reduce the cognitive control demand (*Chung et al., 2021*; *Levy et al., 2019*; *Park et al., 2015*). While these findings unequivocally demonstrate the salience of relevant stationary room cues to use for avoiding shock and irrelevant arena cues to ignore during active place avoidance, the most compelling evidence of cognitive control comes from recording hippocampal ensemble discharge. Hippocampal ensemble discharge purposefully represents current position using stationary room information when the subject is close to the stationary shock zone and alternatively represents rotating arena information when the mouse is far from the stationary shock zone (*Figure 4*, *Chung et al., 2021*). Despite this evidence from task design, behavioral observations, and direct electrophysiological representational switching as required to directly demonstrate cognitive control, one might still argue that it is logically possible that the active place avoidance task does not require cognitive control and this is why the mPFC lesion did not impair place avoidance of the initial shock zone. We consider such reasoning to be unproductive because it presumes that only tasks that require an intact mPFC can be cognitive control tasks. We nonetheless acknowledge that, for some, we have not provided sufficient evidence that the active place avoidance requires cognitive control.

We assert that the evidence presented here is compelling, and that these findings require rejecting the central-computation hypothesis, which states that the mPFC is essential for the neural computations that are necessary for all cognitive control tasks. The present findings do not rule out a role for mPFC and hippocampal interactions in other cognitive processes (*Guise and Shapiro, 2017*; *Negrón-Oyarzo et al., 2018*; *Spellman et al., 2015*; *Sigurdsson et al., 2010*; *Adhikari et al., 2010*). For example, mPFC may be crucial for processes that rely on the ventral hippocampus and perhaps the direct excitatory mPFC-ventral hippocampus and/or the direct mPFC-dorsal hippocampus excitatory (*Rajasethupathy et al., 2015*) and inhibitory connections (*Hoover and Vertes, 2007*; *Malik*

*et al., 2022*). Nonetheless, the present data favor the local-computation hypothesis because bilateral and unilateral inactivation, as well as numerous other manipulations of the dorsal hippocampus, impair learning, consolidation, and retrieval of the conditioned active place avoidance (*Cimadevilla et al., 2000*; *Cimadevilla et al., 2001*; *Wesierska et al., 2005*; *Jezek et al., 2002*), and optogenetic silencing of dentate gyrus impairs conflict learning. Indeed, we directly confirmed that mPFC lesion does not reduce the goal-directed representational switching of dorsal hippocampus (*Figure 4*) that is a *sine qua non* for cognitive control of spatial information (*Chung et al., 2021*; *van Dijk and Fenton, 2018*; *Kelemen and Fenton, 2010*; *Talbot et al., 2018*; *Kelemen and Fenton, 2016*; *Dvorak et al., 2018*), despite detecting other behavioral and electrophysiological effects of the lesion. We emphasize that observed representational switching is neurobiological evidence of cognitive control; it is not itself cognitive control, although it could be part of transmissive control. Rather, the observed representational switching indicates that cognitive control persists after mPFC lesion; indeed, what a control signal itself should resemble is uncertain, even in primates (*Badre, 2024*; *Badre et al., 2021*). It is especially important that mPFC lesion impaired conflict avoidance of the relocated shock zone even after eight trials during which both sham and lesion animals learned and performed at a behavioral asymptote (*Figure 1F*). We suggest this identifies a key role for the mPFC in cognitive flexibility, specifically the ability to judiciously select between different memories of the location of shock, and that this is demonstrably distinct from judiciously selecting between room- and arena-based classes of spatial information.

Although muscimol inactivation of mPFC did not impair active place avoidance (*Cernotova et al., 2021*), as observed in the present study, it is possible that acute manipulations of the mPFC, such as optogenetic, chemogenetic, or other pharmacological inactivation would lead to different results. In which case, post-lesion reorganization of neural circuits is sufficient to compensate for the loss of the mPFC functions that support cognitive control as required by the active place avoidance task. However, this would not change the conclusion to reject the central-computation hypothesis. It is also possible that the role of the mPFC is best evaluated in tasks that rely on egocentric spatial information (*Kesner et al., 1989*), rather than the allocentric spatial information upon which the present active place avoidance task variant relies, but this possibility would also require rejecting, or at least severely limiting, the central-computation hypothesis. Finally, we observed impaired performance characterized by cognitive flexibility in the conflict task variant, demonstrating not only that the lesion was effective, but a dissociation of the neural substrates needed for different components of cognitive control, as has been previously reported (*Keeler and Robbins, 2011*; *Dias et al., 1997*). The mPFC has a crucial role in judiciously selecting between distinct memories of the room-frame shock location, which arguably is a form of memory-guided cognitive control that also depends on hippocampus (*van Dijk and Fenton, 2018*; *Dvorak et al., 2018*; *Burghardt et al., 2012*), but not a form of cognitive control that has been demonstrated in the representational discharge of hippocampal cells (*van Dijk and Fenton, 2018*).

Taken together, the findings are consistent with the local-computation hypothesis that diverse (but probably not all) brain circuits can perform the computations needed for cognitive control of the information and/or behavior for which that circuit is specialized to process.

## Materials and methods

All methods complied with the Public Health and Service Policy on Humane Care and Use of Laboratory Animals and were approved by NYU's University Animal Welfare Committee under protocol 12-1383, which follow National Institutes of Health guidelines.

### Subjects

Thirty adult, male, Long–Evans rats were purchased from Charles River to arrive at New York University at approximately 40 days old. The rats were given at least 1 week to acclimate to the facility and were housed two per cage on a 12:12 light:dark cycle with free access to food and water. Fifteen rats were randomly assigned to the lesion group and 14 to the sham group. The experimenters working with the rats were blind to group identity. One animal was used to confirm the coordinates, by injecting fluorogold (Fluorochrome, Denver, CO, 2.5% w/v in distilled water) at the same rate and for the same duration as for the lesions. Three of the sham and three of the mPFC lesions rats were used for the

electrophysiology recordings. The behavior of these rats was not included in the behavioral assessment due to the requirement for extended training sessions to record neuronal activity.

## Behavioral analysis followed by cytochrome oxidase activity

One rat in the lesion group was excluded because the lesion was inadequate and one had to be removed from the study due to a skin irritation, resulting in 10 rats in the lesion group. All of these rats underwent behavioral assessment (lesion n=10), eight of which were stained for CO. The tissue from one lesion and one sham rat could not be used for CO because the tissues were lost to thawing and tissue from one rat from each group was not processed. For the control group, one rat was excluded from performing behavioral experiments because his lesioned cage mate had a skin irritation that resulted in a large skin lesion and required euthanasia. Because the abrupt change to single housing is stressful and can affect cognitive behavior, the control rat was also excluded from behavioral testing; this control brain was still processed for CO activity. One rat was excluded from the behavioral experiments due to equipment malfunction during training, and one rat had tissue damage during the sham surgery that appeared as a lesion, so this rat was excluded from the study. Thus, the sham group had eight rats for behavioral assessment and eight for CO assessment. Our final group numbers for the behavioral assessment were therefore sham group n=8, lesion group n=10, and our final group numbers for CO activity were sham group n=8 and lesion group n=8.

## Lesion surgery

Between the ages of 48–56 days, the rat received PFC or sham lesion under sodium pentobarbital (50 mg/kg, i.p.) anesthesia. Bilateral lesion targeted two sites in each hemisphere at the following stereotaxic coordinates: from Bregma, Site 1: A.P +2.5, ML ± 0.6, DV - 5.0, relative to the skull surface, Site 2: AP +3.5, ML ± 0.6, DV - 5.2, relative to skull surface. Ibotenic acid (0.06 M in PBS) was used to create an excitotoxic lesion following published work (*Birrell and Brown, 2000*). A 0.2 µl volume was injected at each site with a micro-infusion pump at a flow rate 0.2 µl/min through a stainless-steel cannula (0.25 mm outer diameter); the cannula was left in place for 4 min (for a total of 5 min per injection site) before being slowly withdrawn. Rats in the sham group underwent the same surgery procedure, but only PBS was injected. The rats were given 1 week to recover.

## Active place avoidance task

A commercial rotating arena and tracking software (Tracker, Bio-Signal Group, Acton, MA) was used for the active place avoidance task, which has been described previously (*Pavlowsky et al., 2017*; *Pastalkova et al., 2006*; *O'Reilly et al., 2016*). Briefly, the rats were placed on a stainless steel 82 cm diameter arena that could rotate about its center at 1 rpm under computer control. The arena was in the center of a 3 m × 3 m space that was surrounded by black curtains with a distinctive orienting cue (striped fabric). The arena floor was made of parallel stainless-steel rods organized into five electrical poles. A constant current source could be triggered by the software to deliver shock that was scrambled across the five poles to ensure it was unlikely that shock could be avoided by a fortuitous posture or by feces shorting the bars. A 50 cm high transparent PETG wall ensured the rats remained on the arena and allowed them to see into the room as the arena rotated. The position of the rat in the spatial frame of the room was tracked 30 times a second at 3.2 mm resolution by analysis of the video image from an overhead camera. An infrared LED that rotated with the arena was tracked similarly from the video image, and the rat's arena-frame position on the arena surface was computed relative to the arena LED (*Fenton et al., 1998*; *Bures et al., 1998*). The room-frame and arena-frame position time series, along with the delivery of shock, were stored in data files for off-line automated software analysis of end-point measures (TrackAnalysis, Bio-Signal Group, Acton, MA). Three end-point measures are reported. (i) The distance walked measures how much the rat walked on the arena surface. (ii) The time to first enter the shock zone increased with training; it estimates the between-session active place avoidance memory by estimating how well the rat can remember the previous room-frame locations of shock. (iii) The number of entrances into the shock zone decreased with training; it estimates place learning and can decrease due to between-session memory as well as within-session memory of the locations of shock.

## Active place avoidance training

After recovery from surgery, the rats were brought to the laboratory in their home cages that were placed on a rack to the side of the experimental room. The rats were each handled for 5 min per day

for 5 days. Behavioral training consisted of 10 min trials with an intertrial interval of 10 min during which the rat was returned to its home cage in the experimental room. On the first day of behavior training, animals had a pretraining session that consisted of two trials during which the rats were allowed to explore the stationary arena to habituate to the environment. The following two days the rats underwent eight training trials per day on the rotating arena, learning to avoid entering a 60° sector shock zone in which the rats received a mild foot shock (500 ms, 60 Hz, 0.4 mA). This shock zone was defined by the stationary cues in the room and was the same for all rats. On the fourth day of the behavioral training, the rats had a single trial with the shock on to test retention of the training. The rats were then trained on days 4 and 5 in a conflict training session in which the rats received eight trials per day with the shock zone located 180° from the initial training position. All trials were 10 min with 10 min intertrial intervals. The total distance walked during the session measured locomotion, the time to first enter the shock zone location measured avoidance memory, and the number of entrances into the shock zone was used to measure place learning. Savings in place learning on the conflict trials when the shock zone was relocated 180° was assessed for each rat by a savings index, which compares the number of entrances during the first 5 min of the first and second days of conflict training, conflict trials 1 and 9, respectively. The difference is normalized by the number of entrances on the first conflict trial and reported as a percentage:

$$Savings\ Index = \frac{\left(Number\ of\ Entrances_{conflict1} - Number\ of\ Entrances_{conflict9}\right)}{Number\ of\ Entrances_{conflict1}} X100$$

## Electrophysiology procedures

The following procedures have been previously described in detail (*Park et al., 2019*). Briefly, rats were anesthetized (pentobarbital 50 mg/kg) and received mPFC or sham lesions as described above. The Neuropixel recording system was used (*Putzeys et al., 2019*). A Neuropixel electrode array was implanted in the brain to traverse Bregma coordinates AP –3.8 mm, ML 2.5 mm for recording dorsal hippocampus and the tip targeted DV 9.3 mm, which allowed recording from the underlying posterior and lateral posterior thalamus. The Neuropixel was stabilized and protected by mounting it in a custom 3-D printed appliance. The appliance and five bone screws were cemented to the skull (Unifast, GC America Inc, IL). The rats were allowed at least 1 week to recover before further manipulation.

The rats received active place avoidance training as described above. In addition, to permit neural ensemble recordings, after the last pretraining, training, and conflict training sessions, shock was turned off and the session was extended for 10–20 min during which CA1 neural ensemble discharge was recorded using SpikeGLX. Signals were filtered between 300 Hz and 10 kHz and sampled at 30 kHz for single unit recording. Because of these extended sessions, the behavior from these animals was not included in our behavioral assessment.

## Electrophysiology data analysis

Single units were automatically sorted using Kilosort 2 (*Pachitariu et al., 2016*). Units were only studied if the estimated contamination rate was <20% with spikes from other neurons. This was estimated from refractory period violations relative to expected. Units with non-characteristic or noisy waveforms were also excluded. We computed a 'burst ratio' to characterize burstiness of discharge as number of spikes with ISI ≤30 ms divided by number of spikes with 100 ms ≥ ISI ≤ 130 ms.

Hippocampus single units were classified as complex-spike or theta cells as in prior work (*Kelemen and Fenton, 2010*; *Fenton et al., 2008*; *Ranck, 1973*). Complex-spike cells appear to be pyramidal cells, having long-duration waveforms (>250 μs), low discharge rate (<5 AP/s), and a tendency to fire in bursts. Theta cells are likely local interneurons (*Fox and Ranck, 1975*) and had short-duration waveforms (<250 μs), high discharge rate (>2 AP/s), and did not tend to fire in bursts. Only hippocampal units classified as pyramidal cells were studied. Thalamic units, recorded from the posterior and lateral posterior thalamic nucleus, which lies directly beneath the hippocampus, fired at high rates and were less likely to discharge in bursts (*Supplementary file 3*).

The representational dynamics of CA1 discharge alternating between preferentially signaling the current location in the stationary room or the current location on the rotating arena was investigated to directly evaluate cognitive control during place avoidance behavior. Spatial-frame-specific

momentary positional information ($I_{pos}$) is used to estimate the location-specific information from the activity of a cell during a brief interval (Δt=133 ms).

$$I_{\text{pos}}(t) = p_{i|x} \log_2 \left( \frac{p_{i|x}}{p_i} \right), p_{i|x}$$ is the probability of observing i activity at location x, and $p_i$

is the overall probability of observing i activity. $I_{pos(room)}$ estimates the information about the current location x in the room, whereas $I_{pos(arena)}$ separately estimates the information about current location x on the rotating arena (*Olypher et al., 2003*). $I_{\text{pos}}(t)$ can be positive or negative, but the absolute value is large whenever the cell's activity at the current location is distinct or 'surprising' compared to the location-independent probability of observing the same activity. The value $|I_{pos}(t)|$ is abbreviated $I_{pos}$. A unit's spatial-frame preference is estimated at each Δt moment as the difference:

$\Delta I_{\text{pos}} = I_{\text{pos}}(\text{room}) - I_{\text{pos}}(\text{arena})$, where positive values indicate a momentary preference for signaling the room location, and negative values indicate a preference for signaling the arena location. $\Delta I_{pos}$ time series were evaluated for significance compared to chance fluctuations around the mean using the z value from a runs test. The SFEP at each Δt was computed as the average $\Delta I_{pos}$ over all cells. This ensemble $\Delta I_{pos}$ time series (assessed for significance by runs test) was averaged over all times and locations to estimate an overall SFEP as the proportion of time the ensemble activity was room (or arena) preferring (*Chung et al., 2021*; *van Dijk and Fenton, 2018*). To evaluate if SFEP was purposeful, the arena was divided into approximately two halves based on where individual rats avoided during a retention test with shock off. The 'near' half was the largest sector that includes at least 50% of the recording time and surrounds the least visited 20° sector of the room frame (within the shock zone). The 'far' half is the remaining sector. The probability of observing room-preferring $\Delta I_{pos}$ was computed at each location to make a representational preference map and the average probability in the near and far halves were compared by statistical evaluation.

Place cell overdispersion measures discharge non-stationarity by estimating how much neuronal discharge deviates from what is expected if the discharge only signals position (*Fenton and Muller, 1998*). Increased overdispersion indicates an extra-positional signal like attention or cognitive control is modulating discharge up and down in addition to position, whereas decreased overdispersion indicates such modulation is dampened, possibly to sustained attention (*Fenton et al., 2010*). Overdispersion of hippocampal discharge was computed as previously described (*Fenton et al., 2010*). Briefly, The spike and position time series is divided into 5 s episodes and the expected firing (*exp*) is computed assuming an inhomogeneous Poisson process. The standardized rate is computed as the normalized difference between the observed (obs) and expected firing: $z = \frac{obs-exp}{\sqrt{exp}}$. Episodes with exp =0 are undefined, and only episodes in which *exp* is greater than the cell's mean firing were analyzed because this criterion selects episodes during which the rat passes through the central region of the firing field. The variance of the distribution of z values quantifies overdispersion, which can be compared between conditions using a F test.

## Tissue processing and histochemistry
### Verification of lesion coordinates

Immediately following surgery to inject fluorogold, the rat was transcardially perfused with 0.9% saline and 10% formalin, and the brain extracted and postfixed in 10% formalin for 24 h at 4°C. The brain was then cryoprotected in 30% sucrose (w/v in 1× phosphate-buffered saline) and stored at 4°C until cut on a cryostat (40 µm). The sections were mounted onto gelatin-coated slides and scanned using an Olympus VS120 microscope (fluorescence, ×10). The slides were then placed in e-pure water (two times, 1 min each) before being dehydrated (50, 70, 80, 95, 100, 100% ethanol, 1 min each) and defatted in 50:50 chloroform ethanol for 20 min. The slides were then rehydrated (100, 100, 90, 80, 70, 50% ethanol, 1 min each) before being Nissl stained with Cresyl violet (1 min), rinsed in e-pure water, and dehydrated (50% ethanol, 1 min, 70% ethanol until the white matter was white, 95% ethanol with acetic acid until desired intensity, 100% ethanol, 2 min each). The tissues were then cleared in xylenes (three times, 5 min each) before being coverslipped. The slides were then scanned again (light microscope, ×10). Images were manipulated using Adobe Photoshop CS6 to perform auto-contrast on each image and to remove background from the images. The fluorescent images were then all thresholded in a single operation to remove the background, and the result was superimposed on the corresponding Nissl-stained sections using Adobe Illustrator.

## Cytochrome oxidase activity and Nissl staining

On the day following the completion of behavior training, the rats were anesthetized with isoflurane, immediately decapitated, and the brains were extracted. The brains were rapidly frozen in isopentane on dry ice and stored at –80°C. Sets of brains consisting of 2–4 animals per group were cut simultaneously on a cryostat (40 µm), and sorted into the three series, one of which was Nissl stained and one used for CO histochemistry. The series used for Nissl staining was stored at room temperature and the other two series stored at –80°C until processed for CO histochemistry. To control for variability across batches of histochemical staining, 20, 40, 60, and 80 µm sections of fresh rat brain tissue homogenate (prepared as in *Shumake et al., 2000*) were included. CO staining was performed according to *O'Reilly et al., 2009*. Stained slides were scanned with an Olympus VS 120 light microscope (2×) and the optical densities measured from captured images. Images were converted to 8-bit gray scale using ImageJ (*Schneider et al., 2012*) and optical densities read from the standard slides and 14 brain regions (*Figure 1*). These brain regions included the dysgranular and granular retrosplenial cortices (RSD and RSG, respectively), the nucleus reuniens (RE), the central nucleus of the amygdala (CEA), basomedial and basolateral amgydala (BMA, and BLA, respectively), the dorsal hippocampal CA1, CA2, CA3 and dentate gyrus areas (dCA1, dCA2, dCA3, dDG, respectively), the ventral hippocampal CA1, CA3 and dentate gyrus areas (vCA1,vCA3, vDG, respectively), and the dorsal subiculum (DS). The optical densities were measured using ImageJ (NIH) and CO activity was normalized as in *O'Reilly et al., 2016*. Optical densities were measured while blind to the group identity. Three to six optical density readings were taken for each brain region, from both hemispheres and averaged for each individual subject.

To confirm the lesion site, one series of sections was stained with Cresyl violet. The slides were placed in the e-pure water (two times, 1 min each) and dehydrated in a series of ethanol baths (50, 70, 80, 90, 100, 100%, 1 min each) prior to clearing the fats in a 1:1 mixture of ethanol:chloroform for 20 min. The slides were then rehydrated (100, 100, 90, 80, 70, 50% ethanol, 1 min each), rinsed in e-pure water, and placed in the Cresyl violet for 10 min. The slides were again dehydrated and cleared in xylenes (three times, 5 min each) before being coverslipped. Images were captured with an Olympus VS120 light microscope at 10×. Contours of the lesion area were drawn using Adobe Illustrator in a layer overlaid on the tissue images. The images were matched to the appropriate section of the Paxinos and Watson rat brain atlas (*Paxinos and Watson, 2007*). The filled contours were converted to a .tiff file for quantification of the lesion area (pixels) using ImageJ (see *Figure 1—figure supplement 2*). Rats with verified bilateral lesions that spanned at least three sections were included in the data analysis; one rat was excluded from analysis because the lesion was only evident in one hemisphere.

## Statistical analysis

### Cytochrome oxidase activity

Group averages of the optical densities were calculated for each brain region and expressed as mean ± SEM relative CO activity/µm of tissue. Interregional metabolic covariation was examined by calculating Pearson correlations between each brain region. The statistical significance of interregional CO activity correlations was determined by a 0.01 false discovery rate. To determine if the interregional correlations were significantly different between groups, we transformed the r value of the correlation to Fisher's z-scores. Significant correlations ($p < 0.05$) were used to generate graph theoretical networks using the Brain Connectivity Toolbox in MATLAB (https://sites.google.com/site/bctnet/).

### Behavior and electrophysiology

Place avoidance task performance and SFEP electrophysiology measures were compared using multivariate analysis of variance. Repeated measure analyses were performed using the Greenhouse–Geisser correction. One-factor group comparisons were performed by *t*-test. Pairs of binomial proportions were compared by a test for proportions (z). Runs tests were used to evaluate the likelihood of observing the ensemble $\Delta I_{pos}$ time series by chance. Statistical significance was set at 0.05 for all comparisons. Test statistics, degrees of freedom, and effect sizes are provided in addition to exact p values. These values were computed using SPSS Statistics 28 and Microsoft Excel, and sample sizes were based on effect sizes in prior work and computed using G*Power version 3.1.9.6.

## Acknowledgements

Supported by NIH grants R01NS105472, R01MH115304, and R01MH132204 to AAF.

## Additional information

### Funding

| Funder | Grant reference number | Author |
| --- | --- | --- |
| National Institute of Neurological Disorders and Stroke | R01NS105472 | André A Fenton |
| National Institute of Mental Health | R01MH115304 | André A Fenton |
| National Institute of Mental Health | R01MH132204 | André A Fenton |

The funders had no role in study design, data collection and interpretation, or the decision to submit the work for publication.

### Author contributions

Eun Hye Park, Conceptualization, Data curation, Formal analysis, Investigation, Visualization, Writing – review and editing; Kally C O'Reilly Sparks, Conceptualization, Formal analysis, Investigation, Methodology, Visualization, Writing – review and editing; Griffin Grubbs, David Taborga, Kyndall Nicholas, Armaan S Ahmed, Natalie Ruiz-Péreza, Natalie Kim, Investigation; Simon Segura-Carrillo, Formal analysis, Validation, Visualization; André A Fenton, Conceptualization, Formal analysis, Funding acquisition, Methodology, Project administration, Resources, Supervision, Writing – original draft

### Author ORCIDs

Eun Hye Park ⓘ https://orcid.org/0000-0001-9180-7579
Simon Segura-Carrillo ⓘ https://orcid.org/0000-0001-8287-5158
André A Fenton ⓘ https://orcid.org/0000-0002-5063-1156

### Ethics

Every effort was made to minimize suffering. All methods complied with the Public Health and Service Policy on Humane Care and Use of Laboratory Animals and were approved by NYU's University Animal Welfare Committee under protocol 12-1383. All surgery was performed under anesthesia with analgesia during recovery.

Reviewer #1 (Public review): https://doi.org/10.7554/eLife.104475.3.sa1
Reviewer #2 (Public review): https://doi.org/10.7554/eLife.104475.3.sa2
Reviewer #3 (Public review): https://doi.org/10.7554/eLife.104475.3.sa3
Author response https://doi.org/10.7554/eLife.104475.3.sa4

## Additional files

### Supplementary files

Supplementary file 1. Average cytochrome oxidase activity in sham and lesioned brains. Relative cytochrome oxidase (CO) activity/µm tissue ($\times 10^{-1}$). RSD, dysgranular retrosplenial cortex; RSG, granular retrosplenial cortex; RE, the nucleus reuniens; CEA, the central nucleus of the amygdala; BMA, basomedial amygdala; BLA, basolateral amygdala; dCA1, dorsal CA1; dCA2, dorsal CA2; dCA3, dorsal CA3; dDG, dorsal dentate gyrus; vCA1, ventral CA1; vCA3, ventral CA3; vDG, ventral dentate gyrus; DS, dorsal subiculum (sham; n=8, lesion; n=8).

Supplementary file 2. Place cell counts and behavioral measures of the recorded rats.

Supplementary file 3. Electrophysiological properties of thalamic cells.

MDAR checklist

## Data availability

The electrophysiological and behavioral tracking data are available as a dataset at Harvard Dataverse: https://doi.org/10.7910/DVN/D6A2H0.

The following dataset was generated:

| Author(s) | Year | Dataset title | Dataset URL | Database and Identifier |
|-----------|------|---------------|-------------|------------------------|
| Fenton AA | 2025 | Cognitive control of behavior and hippocampal information processing without medial prefrontal cortex | https://doi.org/10.7910/DVN/D6A2H0 | Harvard Dataverse, 10.7910/DVN/D6A2H0 |

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
