## [Editor Report · eLife Assessment]

This **important** study includes **convincing** evidence to show that behavioral measures and hippocampal representations when animals use task-relevant information and ignore irrelevant information do not depend on the medial prefrontal cortex. The results are expected to be of interest to those studying neural mechanisms of cognitive control and functions of associational brain regions.

---

## [Referee Report · Reviewer #1 (Public review)]

Summary:

The authors examine the role of the medial prefrontal cortex (mPFC) in cognitive control, i.e. the ability to use task-relevant information and ignore irrelevant information, in the rat. According to the central-computation hypothesis, cognitive control in the brain is centralized in the mPFC and according to the local hypothesis, cognitive control is performed in task-related local neural circuits. Using the place avoidance task which involves cognitive control, it is predicted that if mPFC lesions affect learning, this would support the central computation hypothesis whereas no effect of lesions would rather support the local hypothesis. The authors thus examine the effect of mPFC lesions in learning and retention of the place avoidance task. They also look at functional interconnectivity within a large network of areas that could be activated during the task by using cytochrome oxydase, a metabolic marker. In addition, electrophysiological unit recordings of CA1 hippocampal cells are made in a subset of (mPFC-lesioned or intact) animals to evaluate overdispersion, a firing property that reflects cognitive control in the hippocampus. The results indicate that mPFC lesions disrupted correlations of activity between functionally-related regions. Behaviorally, lesions did not impair place avoidance learning and retention (though flexibility was altered during conflict training). In addition, hippocampal place cell overdispersion was decreased in lesioned rats only in the absence of cognitive control challenge (pretraining). Cognitive control seen in hippocampal place cell activity (alternation of frame-specific firing) was not affected by the lesion. Overall, the absence of effects of mPFC lesions on cognitive control in the task or in hippocampal place cells firing support the local hypothesis.

Strengths:

Straightforward hypothesis: clarification of the involvement of the mPFC in the brain is expected and achieved. Appropriate use of fully mastered methods (active place avoidance task, electrophysiological unit recordings, measure of metabolic marker cytochrome oxidase) and rigorous analysis of the data. The conclusion is strongly supported by the data.

Weaknesses:

No notable weaknesses in the conception, making of the study and data analysis.

Comments on revisions:

The authors have satisfactorily addressed all my comments in the revised version.

---

## [Referee Report · Reviewer #2 (Public review)]

Park et al. set out to test two competing hypotheses about the role of the medial prefrontal cortex (PFC) in cognitive control, the ability to use task-relevant cues and ignore task-irrelevant cues to guide behavior. The "central computation" hypothesis assumes that cognitive control relies on computations performed by the PFC, which then interacts with other brain regions to accomplish the task. Alternatively, the "local computation" hypothesis suggests that computations necessary for cognitive control are carried out by other brain regions that have been shown to be essential for cognitive control tasks, such as the dorsal hippocampus and the thalamus. If the central computation hypothesis is correct, PFC lesions should disrupt cognitive control. Alternatively, if the local computation hypothesis is correct, cognitive control would be spared after PFC lesions. The task used to assess cognitive control is the active place avoidance task in which rats must avoid a sector of a rotating arena using the stationary room cues and ignoring the local olfactory cues on the rotating platform. Performance on this task has previously been shown to be disrupted by hippocampal lesions and hippocampal ensembles dynamically represent the room and arena depending on the animal's proximity to the shock zone. They found no group (lesion vs. sham) differences in the three behavioral parameters tested: distance traveled, latency to enter the shock zone, and number of shock zone entries for both the standard task and the "conflict" task in which the shock zone was rotated by 180 degrees. The only significant difference was the savings index; the lesion group entered the new shock zone more often than the sham group during the first 5 minutes of the second conflict session. This deficit was interpreted as a cognitive flexibility deficit rather than a cognitive control failure. Next, the authors compared cytochrome oxidase activity between sham and lesion groups in 14 brain regions and found that only the amygdala shows significant elevation in the lesion vs. sham group. Pairwise correlation analysis revealed a striking difference between groups, with many correlations between regions lost in the lesion group between reuniens and hippocampus, reuniens and amygdala and a correlation between dorsal CA1 and central amygdala that appeared in the lesion group and were absent in the sham group. Finally, the authors assessed dorsal hippocampal representations of the spatial frame (arena vs. room) and found no differences between lesion and sham groups. The only difference in hippocampal activity was reduced overdispersion in the lesion group compared to the sham group on the pretraining session only and this difference disappeared after the task began. Collectively, the authors interpret their findings as supporting the local computation hypothesis; computations necessary for cognitive control occur in brain regions other than the PFC.

Strengths:

The data were collected in a rigorous way with experimental blinding and appropriate statistical analyses.

Multiple approaches were used to assess differences between lesion and sham groups, including behavior, metabolic activity in multiple brain regions, and hippocampal single unit recording.

Weaknesses:

Only male rats were used with no justification provided for excluding females from the sample.

The conceptual framework used to interpret the findings was to present two competing hypotheses with mutually exclusive predictions about the impact of PFC lesions on cognitive control. The authors then use mainly null findings as evidence in support of the local computation hypothesis. They acknowledge that some people may question the notion that the active place avoidance task indeed requires cognitive control, but then call the argument "circular" because PFC has to be involved in cognitive control. This assertion does not address the possibility that the active place avoidance task simply does not require cognitive control.

The authors did not link the CO activity with the behavioral parameters even though the CO imaging was done on a subset of the animals that ran the behavioral task nor do they make any attempt to interpret these findings in light of the two competing hypotheses posed in the introduction. Moreover, the discussion is lacking any mechanistic interpretations of the findings. For example, there are no attempts to explain why amygdala activity and its correlation with dCA1 activity might be higher in the PFC lesioned group.

Publishing null results is important to avoid wasting animals, time, and money. This study's results will have a significant impact on how the field views the role of the PFC in cognitive control. Whether or not some people reject the notion that the active place avoidance task measures cognitive control, the findings are solid and can serve as a starting point for generating hypotheses about how brain networks change when deprived of PFC input.

---

## [Referee Report · Reviewer #3 (Public review)]

Summary:

This study by Park and colleagues investigated how the medial prefrontal cortex (mPFC) influences behavior and hippocampal place cell activity during a two-frame active place avoidance task in rats. Rats learned to avoid the location of mild shock within a rotating arena, with the shock zone being defined relative to distal cues in the room. Permanent chemical lesions of the mPFC did not impair the ability to avoid the shock zone by using the distal cues and ignoring proximal cues in the arena. In parallel, hippocampal place cells alternated between two spatial tuning patterns, one anchored to the distal cues and the other to the proximal cues, and this alteration was not affected by the mPFC lesion. Based on these findings, the authors argue that the mPFC is not essential for differentiating between task-relevant and irrelevant information.

Strengths:

This study was built on substantial work by the Fenton lab that validated their two-frame active place avoidance task and provided sound theoretical and analytical foundations. Additionally, the effectiveness of mPFC lesions was validated by several measures, enabling the authors to base their argument on the lack of lesion effects on behavior and place cell dynamics.

Weaknesses:

The authors define cognitive control as "the ability to judiciously use task-relevant information while ignoring salient concurrent information that is currently irrelevant for the task." (Lines 77-78). This definition is much simpler than the one by Miller and Cohen: "the ability to orchestrate thought and action in accordance with internal goals (Ref. 1)" and by Robbins: "processes necessary for optimal scheduling of complex sequence of behaviour." (Dalley et al., 2004, PMID: 15555683). Differentiating between task-relevant and irrelevant information is required in various behavioral tasks, such as differential learning, reversal learning, and set-shifting tasks. Previous rodent behavioral studies have shown that the integrity of the mPFC is necessary for set-shifting but not for differential or reversal learning (e.g., Enomoto et al., 2011, PMID: 21146155; Cho et al., 2015, PMID: 25754826). In the present task design, the initial training is a form of differential learning between proximal and distal cues, and the conflict training is akin to reversal learning. Therefore, the lack of lesion effects is somewhat expected. It would be interesting to test whether mPFC lesions impair set-shifting in their paradigm (e.g., the shock zone initially defined by distal cues and later by proximal cues). If the mPFC lesions do not impair this ability and associated hippocampal place dynamics, it will provide strong support for the authors' local-computation hypothesis.

Comments on revisions:

The authors fully addressed my comments. I do not have any additional suggestions.

---

## [Author Response]

The following is the authors’ response to the original reviews

**Reviewer #1 (Public review):**
Summary:The authors examine the role of the medial prefrontal cortex (mPFC) in cognitive control, i.e. the ability to use task-relevant information and ignore irrelevant information, in the rat. According to the central-computation hypothesis, cognitive control in the brain is centralized in the mPFC and according to the local hypothesis, cognitive control is performed in task-related local neural circuits. Using the place avoidance task which involves cognitive control, it is predicted that if mPFC lesions affect learning, this would support the central computation hypothesis whereas no effect of lesions would rather support the local hypothesis. The authors thus examine the effect of mPFC lesions in learning and retention of the place avoidance task. They also look at functional interconnectivity within a large network of areas that could be activated during the task by using cytochrome oxidase, a metabolic marker. In addition, electrophysiological unit recordings of CA1 hippocampal cells are made in a subset of (lesioned or intact) animals to evaluate overdispersion, a firing property that reflects cognitive control in the hippocampus. The results indicate that mPFC lesions do not impair place avoidance learning and retention (though flexibility is altered during conflict training), do not affect cognitive control seen in hippocampal place cell activity (alternation of frame-specific firing), a measure of location-specific firing variability, in pretraining. It nevertheless has some effect on functional interconnections. The results overall support the local hypothesis.Strengths:Straightforward hypothesis: clarification of the involvement of the mPFC in the brain is expected and achieved. Appropriate use of fully mastered methods (behavioral task, electrophysiological recordings, measure of metabolic marker cytochrome oxidase) and rigorous analysis of the data. The conclusion is strongly supported by the data.Weaknesses:No notable weaknesses in the conception, making of the study, and data analysis. The introduction does not mention important aspects of the work, i.e. cytochrome oxidase measure and electrophysiological recordings. The study is actually richer than expected from the introduction.

The revised Introduction now includes:

“We used cytochrome oxidase, a metabolic marker of baseline neuronal activity, to confirm the mPFC lesions were effective and that there are non-local network consequences despite the local lesion. We first evaluated cytochrome oxidase activity in regions known to be associated with performance in the active place avoidance task, or regions with known connectivity to the mPFC. We then evaluated covariance of activity amongst the regions in an effort to detect network consequences of the lesion.”

**Reviewer #2 (Public review):**
Park et al. set out to test two competing hypotheses about the role of the medial prefrontal cortex (PFC) in cognitive control, the ability to use task-relevant cues and ignore taskirrelevant cues to guide behavior. The "central computation" hypothesis assumes that cognitive control relies on computations performed by the PFC, which then interacts with other brain regions to accomplish the task. Alternatively, the "local computation" hypothesis suggests that computations necessary for cognitive control are carried out by other brain regions that have been shown to be essential for cognitive control tasks, such as the dorsal hippocampus and the thalamus. If the central computation hypothesis is correct, PFC lesions should disrupt cognitive control. Alternatively, if the local computation hypothesis is correct, cognitive control would be spared after PFC lesions. The task used to assess cognitive control is the active place avoidance task in which rats must avoid a section of a rotating arena using the stationary room cues and ignoring the local olfactory cues on the rotating platform. Performance on this task has previously been shown to be disrupted by hippocampal lesions and hippocampal ensembles dynamically represent the room and arena depending on the animal's proximity to the shock zone. They found no group (lesion vs. sham) differences in the three behavioral parameters tested: distance traveled, latency to enter the shock zone, and number of shock zone entries for both the standard task and the "conflict" task in which the shock zone was rotated by 180 degrees. The only significant difference was the savings index; the lesion group entered the new shock zone more often than the sham group during the first 5 minutes of the second conflict session. This deficit was interpreted as a cognitive flexibility deficit rather than a cognitive control failure. Next, the authors compared cytochrome oxidase activity between sham and lesion groups in 14 brain regions and found that only the amygdala showed significant elevation in the lesion vs. sham group. Pairwise correlation analysis revealed a striking difference between groups, with many correlations between regions lost in the lesion group between reuniens and hippocampus, reuniens and amygdala and a correlation between dorsal CA1 and central amygdala that appeared in the lesion group and were absent in the sham group. Finally, the authors assessed dorsal hippocampal representations of the spatial frame (arena vs. room) and found no differences between lesion and sham groups. The only difference in hippocampal activity was reduced overdispersion in the lesion group compared to the sham group on the pretraining session only and this difference disappeared after the task began. Collectively, the authors interpret their findings as supporting the local computation hypothesis; computations necessary for cognitive control occur in brain regions other than the PFC.Strengths:(1) The data were collected in a rigorous way with experimental blinding and appropriate statistical analyses.(2) Multiple approaches were used to assess differences between lesion and sham groups, including behavior, metabolic activity in multiple brain regions, and hippocampal singleunit recording.Weaknesses:(1) Only male rats were used with no justification provided for excluding females from the sample.

This is a weakness we acknowledge. The experiments were performed at a time when we did not have female rats in the lab.

(2) The conceptual framework used to interpret the findings was to present two competing hypotheses with mutually exclusive predictions about the impact of PFC lesions on cognitive control. The authors then use mainly null findings as evidence in support of the local computation hypothesis. They acknowledge that some people may question the notion that the active place avoidance task indeed requires cognitive control, but then call the argument "circular" because PFC has to be involved in cognitive control. This assertion does not address the possibility that the active place avoidance task simply does not require cognitive control.

We beg to differ that the possibility was not addressed. Prior to making the assertion, the manuscript describes the evidence that the active place avoidance task requires cognitive control. The evidence is multifold, and includes task design, behavior, and electrophysiology; we argue that this is more evidence than has been provided for other tasks that are asserted to require cognitive control. Specifically line 417 states:

“We have previously demonstrated cognitive control in the active place avoidance task variant we used (Fig. 1) because the rats must ignore local rotating place cues to avoid the stationary shock zone. Even when the arena does not rotate, rats distinctly learn to avoid the location of shock according to distal visual room cues and local olfactory arena cues, such that the distinct place memories can be independently manipulated using probe trials [49, 50]. When the arena rotates as in the present studies, neural manipulations that impair the place avoidance are no longer impairing when the irrelevant arena cues are hidden by shallow water [14, 15, 51, 52]. Furthermore, persistent hippocampal neural circuit changes caused by active place avoidance training are not detected when shallow water hides the irrelevant arena cues to reduce the cognitive control demand [10, 31, 33]. While these findings unequivocally demonstrate the salience of relevant stationary room cues to use for avoiding shock and irrelevant arena cues to ignore during active place avoidance, the most compelling evidence of cognitive control comes from recording hippocampal ensemble discharge. Hippocampal ensemble discharge purposefully represents current position using stationary room information when the subject is close to the stationary shock zone and alternatively represents rotating arena information when the mouse is far from the stationary shock zone [Fig. 4; 10].”

Line 436, however, acknowledges a fact that will always be true: no matter what anyone opines - until there are universally agreed upon objective criteria, it is logically possible that active place avoidance does not require cognitive control. The revision states: Despite this evidence from task design, behavioral observations, and direct electrophysiological representational switching as required to directly demonstrate cognitive control, one might still argue that it is logically possible that the active place avoidance task does not require cognitive control and this is why the mPFC lesion did not impair place avoidance of the initial shock zone. We consider such reasoning to be unproductive because it presumes that only tasks that require an intact mPFC can be cognitive control tasks. We nonetheless acknowledge that for some, we have not provided sufficient evidence that the active place avoidance requires cognitive control.

“We assert the evidence is compelling, and together these findings require rejecting the central-computation hypothesis that the mPFC is essential for the neural computations that are necessary for all cognitive control tasks.”

(3) The authors did not link the CO activity with the behavioral parameters even though the CO imaging was done on a subset of the animals that ran the behavioral task nor did they make any attempt to interpret these findings in light of the two competing hypotheses posed in the introduction. Moreover, the discussion lacks any mechanistic interpretations of the findings. For example, there are no attempts to explain why amygdala activity and its correlation with dCA1 activity might be higher in the PFC lesioned group.

The CO study was performed to assess the effects of the lesion, as stated on line 262 “Cytochrome oxidase (CO), a sensitive metabolic marker for neuronal function [27], was used to evaluate whether lesion effects were restricted to the mPFC.” Furthermore, as a matter of fact, line 411 states “Thus, CO imaging and electrophysiological evidence identify changes in the brain beyond the directly damaged mPFC area. In particular, the dorsal hippocampus loses the inhibitory input from mPFC [45, 46] and loses the metabolic correlation with the nucleus reuniens, which is thought to be a relay between the mPFC and the dorsal hippocampus [47, 48].”

These CO measures assess baseline metabolic function and so it would be inappropriate to correlate them with the measures of behavior. Because the lesion and control groups do not differ on most measures of behavior, a relationship to CO measures is not expected. Importantly, even if there were differences in correlations between CO activity and behavioral measures, what could they mean? The study was designed to distinguish between two hypotheses, not to determine what CO differences could mean for behavior. As such, it is not at all clear how metabolic consequences of the lesion relate to the two hypotheses being evaluated, and so we consider it inappropriate to speculate. We did examine, and now include, the correlation between lesion size and conflict behavior. The Fig. 1 legend states “Savings was not related to lesion size r = 0.009, p = 0.98. *p < 0.05.”

(4) Publishing null results is important to avoid wasting animals, time, and money. This study's results will have a significant impact on how the field views the role of the PFC in cognitive control. Whether or not some people reject the notion that the active place avoidance task measures cognitive control, the findings are solid and can serve as a starting point for generating hypotheses about how brain networks change when deprived of PFC input.

We thank the reviewer for the acknowledgement.

**Reviewer #3 (Public review):**
Summary:This study by Park and colleagues investigated how the medial prefrontal cortex (mPFC) influences behavior and hippocampal place cell activity during a two-frame active place avoidance task in rats. Rats learned to avoid the location of mild shock within a rotating arena, with the shock zone being defined relative to distal cues in the room. Permanent chemical lesions of the mPFC did not impair the ability to avoid the shock zone by using distal cues and ignoring proximal cues in the arena. In parallel, hippocampal place cells alternated between two spatial tuning patterns, one anchored to the distal cues and the other to the proximal cues, and this alteration was not affected by the mPFC lesion. Based on these findings, the authors argue that the mPFC is not essential for differentiating between task-relevant and irrelevant information.Strengths:This study was built on substantial work by the Fenton lab that validated their two-frame active place avoidance task and provided sound theoretical and analytical foundations. Additionally, the effectiveness of mPFC lesions was validated by several measures, enabling the authors to base their argument on the lack of lesion effects on behavior and place cell dynamics.Weaknesses:The authors define cognitive control as "the ability to judiciously use task-relevant information while ignoring salient concurrent information that is currently irrelevant for the task." (Lines 77-78). This definition is much simpler than the one by Miller and Cohen: "the ability to orchestrate thought and action in accordance with internal goals (Ref. 1)" and by Robbins: "processes necessary for optimal scheduling of complex sequence of behaviour." (Dalley et al., 2004, PMID: 15555683). Differentiating between task-relevant and irrelevant information is required in various behavioral tasks, such as differential learning, reversal learning, and set-shifting tasks. Previous rodent behavioral studies have shown that the integrity of the mPFC is necessary for set-shifting but not for differential or reversal learning (e.g., Enomoto et al., 2011, PMID: 21146155; Cho et al., 2015, PMID: 25754826). In the present task design, the initial training is a form of differential learning between proximal and distal cues, and the conflict training is akin to reversal learning. Therefore, the lack of lesion effects is somewhat expected. It would be interesting to test whether mPFC lesions impair set-shifting in their paradigm (e.g., the shock zone initially defined by distal cues and later by proximal cues). If the mPFC lesions do not impair this ability and associated hippocampal place dynamics, it will provide strong support for the authors' local computation hypothesis.

Thank you for these comments. In addressing them we have provided a significant revision to the manuscript’s Introduction. While authors like those cited by the reviewer have defined cognitive control, those definitions are difficult to test rigorously, as it is almost a matter of opinion whether a subject is displaying “the ability to orchestrate thought and action in accordance with internal goals" or whether they are using "processes necessary for optimal scheduling of complex sequence of behaviour." What would such definitions of cognitive control predict about neuronal activity? We have deliberately used a simple, operational definition of cognitive control because it is physiologically testable. In the revision, starting at line 93, we have provided an excerpt from Miller and Cohen (2001) with discussion. The importance of that work is that it provides explicit neuronal criteria and a means to operationally define cognitive control. As stated on Line 118 “Accordingly, cognitive control would be at work when there is sustained neuronal network representations of task-relevant information that suppresses or gates representations of salient task-irrelevant information in accord with purposeful judicious behavior.”

We used a R+A- task variant in which there is a stationary room-frame shock zone and task irrelevant arena-frame information. A strict correspondence to shift-shifting task design cannot be accomplished with active place avoidance because an A+R- task that requires avoiding an arena-frame shock zone in the absence of a room-frame shock zone can be accomplished trivially if the subject chooses to not move when it is in a place with no shock. However, the R+A+ task variant is readily learned, in which there is both a room-frame and an arena-frame shock zone (see cited work below). This task variant requires the subject to judiciously shift between avoiding the room-frame shock zone using stationary room information and avoiding the arena-frame shock zone using rotating arena information. This R+A+ task variant might meet the reviewer’s criteria for cognitive control. We have recorded hippocampal and entorhinal ensemble activity during the R+A+ task variant and it is very similar to the activity during the R+A- task we used. Nonetheless, future work will investigate the efect of mPFC lesion on the R+A+ task variant.

Cited work:

Fenton AA, Wesierska M, Kaminsky Y, Bures J (1998), Both here and there: simultaneous expression of autonomous spatial memories in rats. Proc Natl Acad Sci U S A 95:11493-11498. Kelemen E, Fenton AA (2010), Dynamic grouping of hippocampal neural activity during cognitive control of two spatial frames. PLoS Biol 8:e1000403.

Burghardt NS, Park EH, Hen R, Fenton AA (2012), Adult-born hippocampal neurons promote cognitive flexibility in mice. Hippocampus 22:1795-1808.

Park EH, Keeley S, Savin C, Ranck JB, Jr., Fenton AA (2019), How the Internally Organized Direction Sense Is Used to Navigate. Neuron 101:1-9.

**Recommendations for the authors:**

**Reviewer #1 (Recommendations for the authors):**
(1) Incorporate the cytochrome oxidase and hippocampal recordings (rationale and hypothesis) in the introduction, explaining how these aspects are relevant to the general question.

We have done this as requested. See lines 159-173 of the revised introduction.

(2) Figure 1C. On Day 4-5 (conflict training) in which the shock zone was relocated 180 deg from the initial location, the behavioral tracks did not show any presence of the rat in this sector (in particular for the lesion example). Figure 4 nevertheless indicates that entrances have been made (which was expected since rats have to know that the shock zone was relocated).

Thanks for pointing this out. The tracks are from the end of the sessions. The labels have been changed to specify which trials the tracks are from.

(3) Figure 1C. The caption is huge as it contains the statistical analyses details. I would prefer to have these details in the text and keep the caption at a "reasonable" length. At the end of the caption (l. 190-191), it would be less confusing the keep the numbering of the training days: replace D1T1 with D2T1 and D2T9 with D3T9.

The statistical details have been relocated to the main text and the numbering updated, as suggested, thank you.

(4) It was not inconsiderable to show that mPFC lesion had some effects in the present task if it were only to validate the effectiveness of the lesion. This brain area has been shown to be important for planning, cognitive flexibility, etc. Indeed the authors found that the saving index was greater in sham than in mPFC rats (overdispersion in hippocampal firing was also reduced in pretraining) and interpreted this result as impaired flexibility. Would an alternative explanation be a memory deficit? I nevertheless expected that impaired flexibility in mPFC rats would be expressed in conflict trials in the form of more entrances in the zone that was initially not associated with shock (at least in the first trials of Day 4). But it appears to not be the case.

A memory deficit is unlikely to explain the difference between the groups on the first trial of Day 5. Memory in the lesion rats was tested multiple times, specifically at the start of each trial (time to first entrance), including on the 24-h retention test, and no deficits were observed. Performance on Day 9 trial 1 is worse in the lesion group than in the controls, but it is not parsimonious to attribute this to a simple memory deficit since 24-h memory was good and similar between lesion and control rats on days 3 and 4, and memory on Day 5 was equally poor in both the lesion and control rats, as measured by time to first entrance.

(5) Material and methods. The injected volume of ibotenic acid should be mentioned.

The volume 0.2 µl was added. See line 531.

(6) The rationale for doing the conflict training session should be indicated somewhere.

The rationale was provided. See lines 204-208.

**Reviewer #2 (Recommendations for the authors):**
(1) Line 132: The text states that all sham rats improved and only 6/10 lesion rats improved is followed by a t-test, which tests the difference between means; it does not compare proportions. Also, what criterion was used to determine if an improvement was seen or not?

The statistical comparison is provided (now lines 230: test of proportions z = 2.3, p = 0.03). Improvement was simply numerically fewer entrances.

(2) Line 138: This is a very long and confusing sentence. Consider revising for clarity.

The sentence (now line 234) was revised.

(3) Figure 1B only includes data from 3 animals. Most published studies show the whole dataset by presenting the largest and smallest lesions.

Supplemental Figure S2 was added with all the lesions depicted and quantified.

(4) Figure 1C suggestion to make the schematic shock zone line up with the shock zone shown for the tracking data.

Graphically, it looks better as drawn as it uses to perspective to depict a three-dimensional structure.

(5) Methods: Clarify if the shock zone location was the same across all rats.

Line 570 states that the shock zone was the same for all rats.

(6) Line 158: "Behavioral tracks" is not clear. Suggest more precise wording.

Reworded to “Tracked room-frame positions” (now line 249)

(7) Line 166: "effect of trial" - should this be the main effect of trial?; "interaction" - should this be "group x trial" interaction?

Reworded (now line 181).

(8) Line 167: "or their interaction" is awkward in the context of the sentence.

Reworded (now line 182).

(9) Line 182: Avoid talking about "trends" as if they are almost significant unless the authors suspect that they did not have sufficient statistical power to detect differences. In that case, a power analysis should be provided.

Removed.

(10) Line 190: "left:...right..." is hard to follow, especially with acronyms like D1T1. Consider revising for clarity.

Revised (now lines 246-248).

(11) Line 195: "effectiveness of the PFC to impair" is unnecessarily verbose.

Reworded (now lines 255-257).

(12) Savings results: There is a lot of variability in the lesion group. It would be interesting to know if the extent of the lesion correlates with savings.

Savings was not related to lesion. See line 259.

(13) Line 300: The thalamic recording results are not reported in the results section (other than appearing in the table). Moreover, there is no detail about which thalamic nucleus these recordings are from.

Lines 411 and 614 provides these details.

(14) Line 312: "no longer impair" contains a grammatical error.

Corrected (now line 422)

(15) Line 325: "was not impairing" contains a grammatical error.

Corrected (now line 437).

(16) Line 327: The sentence ending with "...opinion of others" seems unnecessarily confrontational.

Previous reviewers at other journals have maintained this position, we therefore included such a strong statement in our initial submission. However, we now revised this statement to avoid appearing confrontational.

(17) Line 329: Sentence is awkward. Consider revising.

Revised (now line 443).

(18) Line 384: The authors should disclose if there was an objective metric for determining the adequacy of the lesion.

The lesion assessment and quantification is better explained in the Methods under “Cytochrome oxidase activity and Nissl staining,” (lines 708-714).

(19) Line 385: The authors should clarify how they got from 15 rats (Line 376) to 10.

This information is provided in the methods.

(20) Line 390: It is not clear why skin irritation in the cage mate would prevent the rat from being tested.

This has been explained in the Methods under “Behavioral analysis followed by cytochrome oxidase activity” (lines 515-518).

(21) Methods section: The authors should describe how the tracking data were acquired. Overhead camera? Tracker based on luminance or body position? What software program was used? What was the sampling rate?

This is now better explained in the Methods under “Active place avoidance task” (lines 538551).

(22) Methods section: Include how fast the arena was rotating and other details about the task such as where rats were placed during the ITI.

Better explained in the Methods under “Active place avoidance task”.

(23) Line 439: The recording system used (hardware & software) should be stated.

This is now included in the Methods (line 538).

(24) Line 435: Though overdispersion calculation is described thoroughly, there is nothing in the paper that tells me what overdispersion means.

What the measure means is now described in the Methods under “Electrophysiology data analysis” (lines 646-650).

(25) Line 561: The test used to assess effect sizes should be stated.

Effect sizes corresponding to the statistical tests are provided.

**Reviewer #3 (Recommendations for the authors):**
(1) At the end of the conflict training, rats with mPFC lesions learned to avoid the new shock zone (Figure 1F, Block 16), but their place cells did not show room-preferring activity near the shock zone (Figure 4B). This observation questions whether spatial frame-specific representation is relevant for active avoidance. Can the authors clarify this point?

This is a dynamic behavior and the hippocampal dynamics match, changing with a dynamic that is a few seconds, as we have shown in several published papers. The lack of a preference averaged over 20 minutes when the rats are avoiding both the current and former shock zones during the conflict session is pretty much what would be expected from such a coarse measurement. The important measure is the spatially-resolved measure of room versus arena preference. Figure 4B shows that in the lesion rats there is less of a frame preference during conflict, generally (consistent with poorer flexibility). However, Figure 4D quantifies the frame preference near and far from the shock zone and accordingly, there is no difference between the groups.

(2) Related to the point above, the author might consider including panels in Figures 4C and D to show the neural activity during the pretraining and conflict training retention period. I assume p(room) will be comparable between the Near and Far segment in both sessions, but the p(room) may be higher in the Conflict training session than the Pretraining session. This would show that the mPFC lesion impairs suppressing the place cell activity encoding the old shock location.

Thanks for the suggestion. While we don’t think we can draw any strong conclusions from this analysis we are fine to show it. The issue is that during conflict, the rats have two perfectly reasonable representations of where there was shock, the initial location that was turned off to make the conflict, and the most recent conflict location of shock. Importantly, these recordings are during conflict retention after we turned off the shock for the retention recording (for the second time in the rat’s experience). Turning off the shock allows us to exactly match the physical conditions of pretraining, initial retention and conflict retention, which was the experimental design’s goal. However, the experiential history of the rats prior to initial retention and conflict retention cannot match, because during initial retention the rats had never experienced a changed shock zone whereas, by conflict retention, they had experienced multiple changes. Importantly, we have previously shown that mouse hippocampal ensembles represent both initial and conflict shock locations, as the animals consider their options during conflict trials (see Dvorak et al 2018, PLoS Biol 16:e2003354). Consequently, we cannot make any strong predictions about whether or not hippocampal activity during conflict retention should be room-frame preferring selectively in the vicinity of the current shock zone. As I am sure the reviewer appreciates from their own introspection, mental representations are mercifully not obliged to dictate behavior. In fact, that is what is interesting and controversial about cognitive control – it is a dynamic internal process and the innovation of our work lies in demonstrating that one cannot only rely on behavior to assess this process. Nonetheless, we did this analysis and now present it in the revised Fig. 4. During pretraining both lesion and sham groups express no particular spatially-modulated preference for either the room or the arena frame, as expected. During initial training both groups express a room-frame preference in the vicinity of the shock zone, as we initially reported. By inspection, during conflict, the sham rats express a preference for room-frame activity in the vicinity of the most recent shock zone location; this preference is weaker than what is expressed during initial retention. The lesion rats do not show this preference. These impressions are quantified in revised Fig. 4D; the comparisons within the conflict retention sessions did not reach statistical significance. We leave it to the reader to interpret what that means. Thanks for the nudge.

(3) The significant group difference in place cell overdispersion during the pretraining phase (Figure 3C) is interesting, but some readers would appreciate additional sentences on its functional implication. Does it mean the spatial tuning of place cells was disrupted by the mPFC lesion?

Only the reliability of spatial firing was altered, not the spatial tuning.

(4) Although the method section described how to calculate overdispersion and SFEP, some concise, intuitive descriptions of these measures in the result section would help readers understand these results.

Overdispersion is better explained. See lines 646-650.

(5) I recommend adding a figure of the task performance of the rats used in the electrophysiological recording experiment and a table summarizing the number of cells recorded per animal.

We have included Table S2 with the cell counts and a summary of the performance for each of the rat in the electrophysiological recording experiment.

(6) Readers would appreciate additional information on task apparatus, such as the size, appearance, and rotating speed of the arena, as well as stationary cues available in the room.

This is now provided in the Methods under “Active place avoidance task”.

(7) Lines 425-416: "On the fourth day of the behavioral training, the rats had a single trial with the shock on to test retention of the training." Shouldn't it be "shock off"?

No the shock was on to prevent extinction learning and to increase the challenge for conflict learning.